# On Singular Distributions With Statistical Structure

**Paul Popescu** [1] **, Vladimir Rovenski** [2,*] **and Sergey Stepanov** [3]

[1]    Department of Applied Mathematics, University of Craiova, Str. Al. Cuza, No, 13, 200585 Craiova, Romania; paul_p_popescu@yahoo.com
[2]    Department of Mathematics, University of Haifa, Mount Carmel, Haifa 31905, Israel
[3]    Department of Mathematics, Finance University, 49-55, Leningradsky Prospect, 125468 Moscow, Russia; s.e.stepanov@mail.ru
[*]    Correspondence: vrovenski@univ.haifa.ac.il

**Abstract:** In this paper, we extend our previous study regarding a Riemannian manifold endowed with a singular (or regular) distribution, generalizing Bochner's technique and a statistical structure. Following the construction of an almost Lie algebroid, we define the central concept of the paper: The Weitzenböck type curvature operator on tensors, prove the Bochner–Weitzenböck type formula and obtain some vanishing results about the null space of the Hodge type Laplacian on a distribution.

**Keywords:** Riemannian manifold; almost Lie algebroid; singular distribution; statistical structure; Weitzenböck curvature operator; harmonic differential form

## 1. Introduction

Distributions (subbundles of the tangent bundle) on a manifold are used to build up notions of integrability, and specifically, of a foliation, e.g., [1–3]. There is definite interest of pure and applied mathematicians to singular distributions and foliations, i.e., having varying dimension, e.g., [4,5]. Another popular mathematical concept is a statistical structure, i.e., a Riemannian manifold endowed with a torsionless linear connection $\widetilde{\nabla}$ such that the tensor $\widetilde{\nabla} g$ is symmetric in all its entries, e.g., [6–12]. The theory of affine hypersurfaces in $\mathbb{R}^{n+1}$ is a natural source of such manifolds; they also find applications in theory of probability and statistics as well as in information geometry.

Recall (e.g., [13]) that a *singular distribution* $\mathcal{D}$ on a manifold $M$ assigns to each point $x \in M$ a linear subspace $\mathcal{D}_x$ of the tangent space $T_x M$ in such a way that, for any $v \in \mathcal{D}_x$, there exists a smooth vector field $V$ defined in a neighborhood $U$ of $x$ and such that $V(x) = v$ and $V(y) \in \mathcal{D}_y$ for all $y$ of $U$. A priori, the dimension of $\mathcal{D}_x$ is not constant and depends on $x \in M$. If $\dim \mathcal{D}_x = \text{const}$, then $\mathcal{D}$ is regular. Singular foliations are defined as families of maximal integral submanifolds (leaves) of integrable singular distributions (certainly, regular foliations correspond to integrable regular distributions). Singular distributions also arise when considering irregular mappings of manifolds, since at the point where the rank of the mapping is less than the dimension of the manifold—the inverse image, the kernel of the mapping arises. Its dimension can vary from point to point. Therefore, the theory presented in the article has applications to differential topology and mathematical analysis.

Let $M$ be a connected smooth $n$-dimensional manifold, $TM$—the tangent bundle, $\mathfrak{X}_M$—the Lie algebra of smooth vector fields on $M$, and $\text{End}(TM)$—the space of all smooth endomorphisms of $TM$. Let $g = \langle \cdot, \cdot \rangle$ be a Riemannian metric on $M$ and $\nabla$—the Levi–Civita connection of $g$.

In this paper, we apply the almost Lie algebroid structure (see a short survey in Section 8) to singular distributions on $M$, and in the rest of paper assume $\mathcal{E} = TM$ and $\rho = P \in \text{End}(TM)$.

**Definition 1** (see [14]). *An image $\mathcal{D} = P(TM)$ of $TM$ under a smooth endomorphism $P \in \mathrm{End}(TM)$ will be called a generalized vector subbundle of $TM$ or a singular distribution.*

**Example 1.** *(a) Let $P \in \mathrm{End}(TM)$ on $(M, g)$ be of constant rank, $0 < r(P) < \dim M$, satisfying*

$$P^2 = P, \qquad P^* = P,$$

*where $P^*$ is adjoint endomorphism to $P$, i.e., $\langle P^*X, Y \rangle = \langle X, PY \rangle$, then we have an almost product structure on $(M, g)$, see [3]. In this case, $P$ and $H = \mathrm{id} - P$ are orthoprojectors onto vertical distribution $P(TM)$ and horizontal distribution $H(TM)$, which are complementary orthogonal and regular, but none of which is in general integrable. Many popular geometrical structures belong to the case of almost product structure, e.g., $f$-structure (i.e., $f^3 + f = 0$) and para-$f$-structure (i.e., $f^3 - f = 0$); such structures on singular distributions were considered in [13]. Almost product structures on statistical manifolds $(M, g, \widetilde{\nabla})$ were studied in [11,12].*

*(b) Let $\mathcal{F}$ be a singular Riemannian foliation of $(M, g)$, i.e., the leaves are smooth, connected, locally equidistant submanifolds of M. e.g., [5]. Then $T\mathcal{F}$ is a singular distribution parameterized by the orthoprojector $P : TM \to T\mathcal{F}$.*

In this article, we generalize Bochner's technique to a Riemannian manifold endowed with a singular (or regular) distribution and a statistical type connection, continue our study [13–18] and generalize some results of other authors in [9]. Recall that the Bochner technique works for skew-symmetric tensors lying in the kernel of the Hodge Laplacian $\Delta_H = d\,\delta + \delta\,d$ on a closed manifold: using maximum principles, one proves that such tensors are parallel, e.g., [19]. Here $d$ is the exterior differential operator, and $\delta$ is its adjoint operator for the $L^2$ inner product. The elliptic differential operator $\Delta_H$ can be decomposed into two terms,

$$\Delta_H = \nabla^*\nabla + \Re, \tag{1}$$

one is the Bochner Laplacian $\nabla^*\nabla$, and the second term (depends linearly on the Riemannian curvature tensor) is called the Weitzenböck curvature operator on $(0, k)$-tensors $S$, e.g., [19].

$$\Re(S)(X_1, \ldots, X_k) = \sum_{a=1}^{k} \sum_{i=1}^{n} (R_{e_i, X_a} S)(\underbrace{X_1, \ldots, e_i, \ldots, X_k}_{a}). \tag{2}$$

Here $\{e_i\}$ is a local orthonormal frame on $(M, g)$ and $\nabla^*$ is the $L^2$-adjoint of the Levi–Civita connection $\nabla$. Note that $\Re$ reduces to Ric when evaluated on (0,1)-tensors, i.e., $k = 1$. According to the well-known formula $(R_{Z,Y} S)(X_1, \ldots, X_k) = -\sum_i S(X_1, \ldots R_{Z,Y} X_i, \ldots, X_k)$ for the action of the curvature tensor $R$ on $(0, k)$-tensors, for $k \geq 2$ the formula from (2) has the form

$$\Re(S)(X_1, \ldots, X_k) = -2 \sum_{i,j,a;b<a} R(e_i, X_a, e_j, X_b) \cdot S(\underbrace{X_1, \ldots, e_j,}_{b} \underbrace{\ldots, e_i,}_{a-b} \ldots, X_k)$$

$$+ \sum_{i,a} \mathrm{Ric}(e_i, X_a) \cdot S(\underbrace{X_1, \ldots, e_i, \ldots, X_k}_{a}),$$

or, in coordinates, $\Re(S)_{i_1, \ldots, i_k} = -2 \sum_{a<b} R_{j\,i_a p\,i_b} S_{i_1 \ldots \,{}^j\, \ldots \,{}^p\, \ldots i_k} + \sum_a \mathrm{Ric}_{i_a j} S_{i_1 \ldots \,{}^j\, \ldots i_k}$. The Weitzenböck decomposition Formula (1) allows us to extend the Hodge Laplacian to arbitrary tensors and is important in the study of interactions between the geometry and topology of manifolds.

Our work has an Introduction section and eight subsequent sections, the References include 25 items. In Section 3, we generalize the notion of statistical structure for the case of distributions. In Sections 2, 4 and 5, following an almost Lie algebroid construction (Section 8 with Appendix) we define the derivatives $\overline{\nabla}^P$ and $\overline{d}^P$, the modified divergence and their $L^2$ adjoint operators on

tensors, and modified Laplacians on tensors and forms. In Section 6, making some assumptions about $P$ (which are trivial when $P = \mathrm{id}_{TM}$), we define the curvature type operator $\overline{R}^P$ of $\overline{\nabla}^P$. In Section 7, we define the Weitzenböck type curvature operator on tensors, prove the Bochner–Weitzenböck type formula and obtain vanishing results. The assumptions that we use are reasonable, as illustrated by examples.

## 2. The Modified Covariant Derivative and Bracket

Here, we define the map $\overline{\nabla}^P : \mathfrak{X}_M \times \mathfrak{X}_M \to \mathfrak{X}_M$ satisfying Koszul conditions, see (48) in Section 8,

$$\overline{\nabla}^P_X Y = \nabla_{PX} Y + K_X Y, \tag{3}$$

called a *P-connection*, which depends on $P$ and a $(1,2)$-tensor $K$ (called *contorsion tensor*), but generally is not a linear connection on $M$. Set $\overline{\nabla}^P_X f = (PX)f$ for $f \in C^1(M)$ (the *P-gradient* of $f$). In particular, for $K = 0$, we have the *P-connection* $\nabla^P$ defined in [13] by

$$\nabla^P_X Y = \nabla_{PX} Y, \tag{4}$$

which plays, in our study, the same role as the Levi–Civita connection in metric-affine geometry. Using $\overline{\nabla}^P$, we construct the *P-derivative* of $(s, k)$-tensor $S$, where $s = 0, 1$, as $(s, k+1)$-tensor $\overline{\nabla}^P S$:

$$(\overline{\nabla}^P S)(Y, X_1, \ldots, X_k) = \overline{\nabla}^P_Y(S(X_1, \ldots, X_k)) - \sum_{i=1}^k S(X_1, \ldots, \overline{\nabla}^P_Y X_i, \ldots, X_k). \tag{5}$$

We use the standard notation $\overline{\nabla}^P_Y S = \overline{\nabla}^P S(Y, \ldots)$. A tensor $S$ is called *P-parallel* if $\overline{\nabla}^P S = 0$.

A linear connection $\widetilde{\nabla} = \nabla + K$ on a Riemannian manifold $(M, g)$ is metric if $\widetilde{\nabla} g = 0$, e.g., [7]; in this case, $K_X^* = -K_X$, where $K_X^*$ is adjoint to $K_X$ with respect to $g$. This concept of metric-affine geometry can be applied for our *P-connections*. Recall that $\nabla^P$ is metric, see [13].

**Proposition 1.** *The P-connection has a metric property, i.e., $\overline{\nabla}^P g = 0$, if and only if the map $K_X \in \mathrm{End}(TM)$, see (3), is skew-symmetric for any $X \in TM$, that is $\langle K_X Y, Z \rangle = -\langle K_X Z, Y \rangle$.*

**Proof.** We calculate using (5),

$$(\overline{\nabla}^P_X g)(Y, Z) = (\nabla_{PX} g)(Y, Z) - \langle K_X Y, Z \rangle - \langle K_X Z, Y \rangle. \tag{6}$$

Since $\nabla$ has the metric property, then $\nabla_{PX} g = 0$, and the claim follows. □

Using (3), define a skew-symmetric *P-bracket* $\overline{[\cdot, \cdot]}_P : \mathfrak{X}_M \times \mathfrak{X}_M \to \mathfrak{X}_M$ by

$$\overline{[X, Y]}_P = \overline{\nabla}^P_X Y - \overline{\nabla}^P_Y X. \tag{7}$$

By (7) and according to definition (49), first formula, in Section 8, the *P-connection* $\overline{\nabla}^P$ is torsion free. According to (47) in Section 8, we use the bracket (7) to define the following operator:

$$\overline{\mathfrak{D}}^P(X, Y) = [PX, PY] - P\overline{[X, Y]}_P.$$

Note that the equality $\overline{\mathfrak{D}}^P = 0$ corresponds to (46), third formula, with $\rho = P$ of a skew-symmetric bracket. The following result generalizes Proposition 3 in [16].

**Proposition 2.** *Condition $\overline{\mathfrak{D}}^P = 0$ is equivalent to the symmetry on covariant components of the $(1,2)$-tensor $\overline{\mathcal{A}}(X, Y) = (\nabla_{PX} P)(Y) - P(K_X Y)$, where $\nabla$ is the Levi–Civita connection of $g$, that is*

$$(\nabla_{PX} P)(Y) - P(K_X Y) = (\nabla_{PY} P)(X) - P(K_Y X). \tag{8}$$

**Proof.** Using (7), we have

$$\overline{[X,Y]}_P = \nabla_{PX}Y - \nabla_{PY}X + K_XY - K_YX. \tag{9}$$

Thus,

$$
\begin{aligned}
\overline{\mathfrak{D}}^P(X,Y) &= \nabla_{PX}PY - P\nabla_{PX}Y - P(K_XY) - \nabla_{PY}PX + P\nabla_{PY}X + P(K_YX) \\
&= \nabla_{PX}PY - \nabla_{PY}PX - P(\overline{\nabla}_X^P Y) + P(\overline{\nabla}_Y^P X) = \overline{\mathcal{A}}(X,Y) - \overline{\mathcal{A}}(Y,X),
\end{aligned}
$$

and the conclusion follows. $\square$

**Theorem 1.** *If (8) holds for a P-connection (3), then the endomorphism P and the bracket $\overline{[\cdot,\cdot]}_P$ given in (7) define an almost algebroid structure on TM.*

**Proof.** This follows from Proposition 2, according to Definition 7 in Section 8. $\square$

**Example 2.** *If $N_P = 0$ (the Nijenhuis tensor of P) and $K = c\,\nabla P$ (where $c \in \mathbb{R}$ and $\nabla$ is the Levi–Civita connection of g), then the tensor $\overline{\mathcal{A}}$ (given in Proposition 2) is symmetric, thus the condition (8) holds.*

### 3. The Statistical *P*-Structure

A linear connection $\widetilde{\nabla}$ on a Riemannian manifold $(M,g)$ is called statistical if it is torsionless and tensor $\widetilde{\nabla}g$ is symmetric in all its entries, e.g., [6,9]. Such a pair $(g,\widetilde{\nabla})$ is called a statistical structure on $M$. In this case,

$$K_X^* = K_X, \quad K_XY = K_YX \quad (X,Y \in TM), \tag{10}$$

equivalently, the statistical cubic form $A(X,Y,Z) = \langle K_XY, Z \rangle$ is symmetric. We generalize this concept for singular distributions.

**Definition 2.** *A P-connection $\overline{\nabla}^P$ on $(M,g)$ will be called statistical if the statistical cubic form $A(X,Y,Z)$ is symmetric, or, equivalently, (10) holds. In this case, the pair $(g,\overline{\nabla}^P)$ is called a statistical P-structure on M.*

**Proposition 3.** *If $\overline{\nabla}^P$ is a statistical P-connection for g then the (3,0)-tensor $\overline{\nabla}^P g$ is symmetric in all its entries, i.e., the following Codazzi type condition holds:*

$$(\overline{\nabla}_X^P g)(Y,Z) = (\overline{\nabla}_Y^P g)(X,Z) = (\overline{\nabla}_X^P g)(Z,Y). \tag{11}$$

**Proof.** The theory of Codazzi tensors is well described in [7]. By (6), (10) and the property $\nabla g = 0$, we have $(\overline{\nabla}_X^P g)(Y,Z) = -2A(X,Y,Z)$, thus all three terms in (11) are equal. $\square$

Since $\nabla_{PX}g = 0$ for the Levi–Civita connection $\nabla$, condition (11) does not impose restrictions on $P$ and it is equivalent to the property "the cubic form $A$ is symmetric".

By (9) and (10), the *P*-bracket of a statistical *P*-structure does not depend on $K$:

$$\overline{[X,Y]}_P = \nabla_{PX}Y - \nabla_{PY}X. \tag{12}$$

If $\overline{\nabla}^P$ is statistical then $\nabla_X^P$, see (4), has the same *P*-bracket and $\mathfrak{D}^P = \overline{\mathfrak{D}}^P$. Proposition 2 yields the following result for a statistical *P*-structure.

**Corollary 1.** *For a statistical P-structure, condition $\overline{\mathfrak{D}}^P = 0$, see (8), is equivalent to*

$$(\nabla_{PX}P)(Y) = (\nabla_{PY}P)(X), \quad X,Y \in \mathcal{X}_M, \tag{13}$$

**Proof.** We can put $\overline{\mathcal{A}}(X,Y) = (\nabla P)(PX,Y)$ and reduce (8) to a simpler view (13). $\square$

The notion of conjugate connection is important for statistical manifolds, see [9,20].

**Definition 3.** *For a P-connection $\overline{\nabla}^P$ on $(M, g)$, its conjugate P-connection $\check{\nabla}^P$ is defined by the following equality:*

$$PX\langle Y, Z\rangle = \langle \overline{\nabla}^P_X Y, Z\rangle + \langle Y, \check{\nabla}^P_X Z\rangle.$$

One may show that $\check{\nabla}^P_X = \nabla^P_X - K^*_X$ holds in general, thus, for a statistical $P$-connection $\overline{\nabla}^P$ the conjugate connection $\check{\nabla}^P$ is given by $\check{\nabla}^P_X = \nabla^P_X - K_X$. In turn, the statistical $P$-connection $\overline{\nabla}^P$ is conjugate to $\check{\nabla}^P$. Note that $\overline{\nabla}^P + \check{\nabla}^P = 2\nabla^P$.

**Remark 1.** *For a conjugate statistical P-connection $\check{\nabla}^P$, we can define the P-bracket by $[\widecheck{X, Y}]_P = \check{\nabla}^P_X Y - \check{\nabla}^P_Y X$ and the tensor $\check{\mathfrak{D}}^P(X, Y) = [PX, PY] - P[\widecheck{X, Y}]_P$. By (10), we have*

$$[\widecheck{\cdot, \cdot}]_P = \overline{[\cdot, \cdot]}_P, \quad \check{\mathcal{A}} = \overline{\mathcal{A}}, \quad \check{\mathfrak{D}}^P = \overline{\mathfrak{D}}^P.$$

From Proposition 3, using Remark 1, we obtain the following corollaries.

**Corollary 2.** *The pairs $(g, \overline{\nabla}^P)$ and $(g, \check{\nabla}^P)$ are simultaneously statistical P-structures on M.*

**Corollary 3.** *A statistical P-structure on $(M, g)$ and its conjugate simultaneously define almost algebroid structures (see definition in Section 8) on TM.*

To simplify the calculations, for the rest of this article we will restrict ourselves to statistical $P$-structures, see (10), and to use the concept of almost Lie algebroid, assume (13).

Define the vector field $E = \sum_i K_{e_i} e_i$. Using (10), we get

$$\langle E, X\rangle = \operatorname{tr}_g K_X, \quad X \in \mathfrak{X}_M.$$

For any $(k + 1)$-form $\omega$, set

$$(K_Y \omega)(X_1, X_2, \ldots, X_k) = -\sum_i \omega(X_1, \ldots, K_Y X_i, \ldots, X_k).$$

Throughout the paper, we use also the operator of *contraction* $\iota_Y$: if $\omega$ is a $k$-form and $Y$ is a vector field, then $\iota_Y \omega$ is a $(k-1)$-form given by $\iota_Y \omega(X_1, \ldots, X_{k-1}) = \omega(Y, X_1, \ldots, X_{k-1})$, where $X_i \in \mathfrak{X}_M$.

**Lemma 1** (see Lemmas 6.2 and 6.3 in [9]). *For any local orthonormal frame $\{e_i\}$ and any k-form $\omega$ we have*

$$\sum_i (K_{e_i} \omega)(e_i, X_2, \ldots, X_k) = -\iota_E \omega(X_1, \ldots, X_k), \tag{14}$$

*and for any $(k + 1)$-form, $k \geq 1$, and an index $a \in \{1, \ldots, k\}$ be fixed, we have*

$$\sum_i \omega(e_i, X_1, \ldots, K_{e_i} X_a, \ldots, X_k) = 0. \tag{15}$$

## 4. The Modified Divergence

Define the *P-divergence* of a vector field $X$ on $(M, g)$ using a local orthonormal frame $\{e_i\}$ by

$$\overline{\operatorname{div}}_P X = \operatorname{trace}(Y \to \overline{\nabla}^P_Y X) = \sum_i \langle \overline{\nabla}^P_{e_i} X, e_i\rangle. \tag{16}$$

The following result on the Stokes Theorem for distributions generalizes Lemma 1 in [13].

**Lemma 2.** *On a Riemannian manifold $(M, g)$ with a statistical P-structure, the condition*

$$(\text{div } P)(X) = \text{tr}_g K_X, \quad X \in \mathcal{X}_M \tag{17}$$

*is equivalent to the following equality:*

$$\overline{\text{div}}_P X = \text{div}(PX), \quad X \in \mathcal{X}_M. \tag{18}$$

**Proof.** Note that

$$\sum_i \langle \nabla_{Pe_i} X, e_i \rangle = \sum_{i,j} \langle Pe_i, e_j \rangle \langle \nabla_{e_j} X, e_i \rangle = \sum_{i,j} \langle e_i, P^* e_j \rangle \langle \nabla_{e_j} X, e_i \rangle$$
$$= \sum_j \langle \nabla_{e_j} X, P^* e_j \rangle = \sum_j \langle P \nabla_{e_j} X, e_j \rangle = \text{div}(PX) - (\text{div } P)(X).$$

Using this, definition (3) and (10), we have

$$\overline{\text{div}}_P X = \sum_i \langle \nabla_{Pe_i} X + K_{e_i} X, e_i \rangle = \text{div}(PX) - (\text{div } P)(X) + \text{tr}_g K_X.$$

From this and (10) the claim follows. □

The following theorem is a direct consequence of Lemma 2.

**Theorem 2.** *Let there be a statistical P-structure on a compact Riemannian manifold $(M, g)$ with boundary satisfies (17). Then for any $X \in \mathcal{X}_M$ we have*

$$\int_M (\overline{\text{div}}_P X) \, d \, \text{vol}_g = \int_{\partial M} \langle X, P(\nu) \rangle \, d\omega,$$

*where, as in the classical case, $\nu$ is the unit inner normal to $\partial M$. In particular, on a Riemannian manifold $(M, g)$ without boundary, for any $X \in \mathcal{X}_M$ with compact support, we have $\int_M (\overline{\text{div}}_P X) \, d \, \text{vol}_g = 0$.*

**Example 3.** *For the tensor $K_X Y = (\text{div } P)(Y) \cdot X$ where $X, Y \in TM$, the property (17) follows from $\text{div } P = 0$. The same holds for a more general (1,2)-tensor $K = c \nabla P$ with any $c \in \mathbb{R}$.*

The following pointwise inner products and norms for $(0, k)$-tensors are used:

$$\langle S_1, S_2 \rangle = \sum_{i_1, \dots, i_k} S_1(e_{i_1}, \dots, e_{i_k}) S_2(e_{i_1}, \dots, e_{i_k}), \quad \|S\| = \sqrt{\langle S, S \rangle}$$

while, for $k$-forms, we set

$$\langle \omega_1, \omega_2 \rangle = \sum_{i_1 < \dots < i_k} \omega_1(e_{i_1}, \dots, e_{i_k}) \omega_2(e_{i_1}, \dots, e_{i_k}).$$

For $L^2$-product of compactly supported tensors on a Riemannian manifold, we set

$$(S_1, S_2)_{L^2} = \int_M \langle S_1, S_2 \rangle \, d \, \text{vol}_g.$$

The following $\overline{\nabla}^{*P}$ maps $(s, k+1)$-tensor, where $s = 0, 1$, to $(s, k)$-tensor:

$$(\overline{\nabla}^{*P} S)(X_1, \dots, X_k) = -\sum_i (\overline{\nabla}^P_{e_i} S)(e_i, X_1, \dots, X_k),$$

and similarly for $\check{\nabla}^{*P}$ and $\nabla^{*P}$. Using (15), we relate $\overline{\nabla}^{*P}$ and $\nabla^{*P}$ for any $k$-form $\omega$:

$$\overline{\nabla}^{*P} \omega = \nabla^{*P} \omega + \iota_E \omega, \quad \check{\nabla}^{*P} \omega = \nabla^{*P} \omega - \iota_E \omega. \tag{19}$$

Thus, $\check{\nabla}^{*P} \omega = \overline{\nabla}^{*P} \omega - 2 \iota_E \omega$.

The "musical" isomorphisms $\sharp : T^*M \to TM$ and $\flat : TM \to T^*M$ are used for rank one tensors, e.g., if $\omega \in T_0^1(M)$ is a 1-form and $X \in \mathfrak{X}_M$ then $\omega(X) = \langle \omega^\sharp, X \rangle = \langle \omega, X^\flat \rangle = X^\flat(\omega^\sharp)$.

The $\overline{\nabla}^{*P}$ is related to the *P-divergence* (16) of $X \in \mathcal{X}_M$ by

$$\overline{\mathrm{div}}_P X = -\overline{\nabla}^{*P} X^\flat. \tag{20}$$

To simplify the calculations and use the results of [13] with $\nabla^P$, we will also consider statistical *P*-structures with stronger conditions than (17),

$$a)\ \mathrm{div}\ P = 0, \qquad b)\ E = 0. \tag{21}$$

In Example 4 in [14] we showed that (21)(a) is reasonable: $\mathrm{div}\ P = 0$ with $P = ff^*$ holds for an *f*-structure with parallelizable kernel if and only if both distributions $f(TM)$ and $\ker f$ are harmonic.

The next result generalizes Proposition 1 in [13] and shows that $\check{\nabla}^{*P}$ is $L^2$-adjoint to $\overline{\nabla}^P$ on *k*-forms.

**Proposition 4.** *If conditions* (21) *hold for a statistical P-connection* $\overline{\nabla}^P$, *then for any compactly supported k-form* $\omega_1$ *and* $k + 1$-form $\omega_2$, *we have*

$$(\check{\nabla}^{*P} \omega_2,\ \omega_1)_{L^2} = (\omega_2,\ \overline{\nabla}^P \omega_1)_{L^2}. \tag{22}$$

**Proof.** Define a compactly supported 1-form $\omega$ by

$$\omega(Y) = \langle \iota_Y \omega_2,\ \omega_1 \rangle, \quad Y \in \mathcal{X}_M.$$

It was shown in Proposition 1 in [13] using assumption $\mathrm{div}\ P = 0$ that

$$-\overline{\nabla}^{*P} \omega = -\langle \overline{\nabla}^{*P} \omega_2,\ \omega_1 \rangle + \langle \omega_2,\ \overline{\nabla}^P \omega_1 \rangle. \tag{23}$$

To simplify further calculations, assume that $k = 1$. Then, using (19) and (23), we obtain

$$-\overline{\nabla}^{*P} \omega = -\langle \check{\nabla}^{*P} \omega_2,\ \omega_1 \rangle + \langle \omega_2,\ \overline{\nabla}^P \omega_1 \rangle + \sum_{i \neq j} \langle \omega_2(e_i, e_j),\ \omega_1(K_{e_i} e_j) \rangle, \tag{24}$$

where $(e_i)$ is a local orthonormal frame on $M$. By symmetry of $K$ and skew-symmetry of $\omega_2$, the last term in (24) vanishes. By (24), (20) and Theorem 2 with $X^\flat = \omega$, we obtain (22). $\square$

The differential operator $\check{\nabla}^{*P} \overline{\nabla}^P$ is called the *P-Bochner Laplacian* for a statistical *P*-structure. The following maximum principle generalizes Proposition 2 in [13].

**Proposition 5.** *Let condition* (17) *hold for a statistical P-connection* $\overline{\nabla}^P$ *on a closed Riemannian manifold* $(M, g)$. *Suppose that* $\omega$ *is a k-form such that* $\langle \check{\nabla}^{*P} \overline{\nabla}^P \omega,\ \omega \rangle \leq 0$. *Then,* $\omega$ *is P-parallel.*

**Proof.** We apply formula (22),

$$0 \geq (\check{\nabla}^{*P} \overline{\nabla}^P \omega,\ \omega)_{L^2} = (\overline{\nabla}^P \omega,\ \overline{\nabla}^P \omega)_{L^2} \geq 0;$$

hence, $\overline{\nabla}^P \omega = 0$. $\square$

## 5. The Modified Hodge Laplacian

Using a statistical *P*-connection $\overline{\nabla}^P$, we define the *exterior P-derivative* of a differential form $\omega \in \Lambda^k(M)$ by

$$\overline{d}^P \omega(X_0, \ldots, X_k) = \sum_i (-1)^i (\overline{\nabla}_{X_i}^P \omega)(X_0, \ldots, X_{i-1}, X_{i+1}, \ldots X_k). \tag{25}$$

For a $k$-form $\omega_p$, the $(k+1)$-form $\overline{\nabla}^P \omega$, see (5),

$$(\overline{\nabla}^P \omega)(Y, X_1, \ldots, X_k) = PY(\omega(X_1, \ldots, X_k)) - \sum_{i=1}^k \omega(X_1, \ldots, \overline{\nabla}^P_Y X_i, \ldots, X_k)$$

is not skew-symmetric, but the form $\overline{d}^P \omega$ is skew-symmetric. For a function $f$ on $M$, we have $\overline{d}^P f = \overline{\nabla}^P f$ and $\check{d}^P f = \check{\nabla}^P f$.

The next proposition (see also Remark 1) generalizes Proposition 5 in [13] and shows that $\check{d}^P = \overline{d}^P$.

**Proposition 6.** *The* $\overline{d}^P : \Omega^k(M) \to \Omega^{k+1}(M)$ *is a 1-degree derivation, see Section 8, that is*

$$\overline{d}^P \omega(X_0, \ldots, X_k) = \sum_{i=0}^k (-1)^i PX_i(\omega(X_0, \ldots, X_{i-1}, X_{i+1}, \ldots, X_k))$$
$$+ \sum_{0 \le i < j \le k} (-1)^{i+j} \omega(\overline{[X, Y]}_P, X_0, \ldots, X_{i-1}, X_{i+1}, \ldots, X_{j-1}, X_{j+1}, \ldots, X_k). \tag{26}$$

**Proof.** This is similar to the proof of Proposition 5 in [13]. For the convenience of a reader we give it here. Using (5) and (25) with $s = 0$, we obtain

$$\overline{d}^P \omega(X_0, \ldots, X_k) = \sum_{i=0}^k (-1)^i PX_i(\omega(X_0, \ldots, X_{i-1}, X_{i+1}, \ldots, X_k))$$
$$+ \sum_{i=0}^k (-1)^i \Big( \sum_{j=0}^{i-1} \omega(X_0, \ldots, \overline{\nabla}^P_{X_i} X_j, \ldots, X_{i-1}, X_{i+1}, \ldots, X_k)$$
$$+ \sum_{j=i+1}^k \omega(X_0, \ldots, X_{i-1}, X_{i+1}, \ldots, \overline{\nabla}^P_{X_i} X_j, \ldots, X_k) \Big)$$
$$= \sum_{i=0}^k (-1)^i PX_i(\omega(X_0, \ldots, X_{i-1}, X_{i+1}, \ldots, X_k))$$
$$+ \sum_{0 \le i < j \le k} (-1)^{i+j} \omega(\overline{\nabla}^P_{X_i} X_j - \overline{\nabla}^P_{X_j} X_i, X_0, \ldots, X_{i-1}, X_{i+1}, \ldots, X_{j-1}, X_{j+1}, \ldots, X_k).$$

Using (7), we complete the proof of (26). $\square$

Put $\overline{\delta}^P = \overline{\nabla}^{*P}$ for the $P$-codifferential $\overline{\delta}^P : \Lambda^k(TM) \to \Lambda^{k-1}(TM)$. Similarly, we define

$$\check{\delta}^P \omega(X_2, \ldots, X_k) = - \sum_i (\check{\nabla}^P_{e_i} \omega)(e_i, X_2, \ldots, X_k).$$

**Proposition 7.** *On a closed* $(M, g)$ *with a statistical P-structure, the P-codifferential* $\check{\delta}^P$ *is* $L^2$*-adjoint to* $\overline{d}^P$*, i.e., for any differential forms* $\omega_1 \in \Lambda^k(TM)$ *and* $\omega_2 \in \Lambda^{k+1}(TM)$ *we have*

$$(\check{\delta}^P \omega_2, \omega_1)_{L^2} = (\omega_2, \overline{d}^P \omega_1)_{L^2}. \tag{27}$$

**Proof.** We derive

$$\langle \overline{d}^P \omega_1, \omega_2 \rangle = \sum_{u=0}^k (-1)^i \overline{\nabla}^P_{\partial_{i_u}} \omega_1(\partial_{i_1}, \ldots, \partial_{i_u}, \ldots, \partial_{i_k}) g^{i_0 j_0} \ldots g^{i_k j_k} \omega_2(\partial_{i_1}, \ldots, \partial_{i_k})$$
$$= (k+1) (\overline{\nabla}^P_{\partial_{i_0}} \omega_1(\partial_{i_1}, \ldots, \partial_{i_k})) g^{i_0 j_0} \ldots g^{i_k j_k} \omega_2(\partial_{j_0}, \ldots, \partial_{j_k}) = \langle \overline{\nabla}^P \omega_1, \omega_2 \rangle,$$

as in the classical case. It appears as a $(k+1)$ factor, that finally is absorbed in the definition of $\overline{d}^P$. Using this and (22), which requires (17), we obtain (27). $\square$

**Definition 4.** *Define the Hodge type Laplacians* $\overline{\Delta}^P_H$ *and* $\check{\Delta}^P_H$ *for differential forms* $\omega$ *by*

$$\overline{\Delta}^P_H \omega = \overline{d}^P \check{\delta}^P \omega + \check{\delta}^P \overline{d}^P \omega, \quad \check{\Delta}^P_H \omega = \overline{d}^P \overline{\delta}^P \omega + \overline{\delta}^P \overline{d}^P \omega. \tag{28}$$

*A differential form* $\omega$ *is said to be P-harmonic if* $\overline{\Delta}^P_H \omega = 0$ *and* $\|\omega\|_{L^2} < \infty$ *(similarly for* $\check{P}$*).*

**Remark 2.** *The P-harmonic forms have similar properties as in the classical case, e.g., (Lemma 9.1.1 in [19]). Let condition (17) hold on a closed $(M, g)$. For $\omega \in \Lambda^k(TM)$, using Proposition 7 and (28), we have*

$$(\overline{\Delta}_H^P \omega, \omega)_{L^2} = (\overline{d}^P \omega, \overline{d}^P \omega)_{L^2} + (\check{\delta}^P \omega, \check{\delta}^P \omega)_{L^2},$$

*thus, $\omega$ is P-harmonic (and similarly for $\check{P}$-harmonic) if and only if $\overline{d}^P \omega = 0$ and $\check{\delta}^P \omega = 0$. Observe that, if $\overline{\Delta}_H^P \omega = 0$ and $\omega = \overline{d}^P \theta$, then $\check{\delta}^P \overline{d}^P \theta = \check{\delta}^P \omega = 0$. It follows that*

$$(\omega, \omega)_{L^2} = (\overline{d}^P \theta, \overline{d}^P \theta)_{L^2} = (\theta, \check{\delta}^P \overline{d}^P \theta)_{L^2} = (\theta, \check{\delta}^P \omega)_{L^2} = 0.$$

*Thus, if $\omega \in \Lambda^k(TM)$ is P-harmonic and $\omega = \overline{d}^P \theta$ for some $\theta \in \Lambda^{k-1}(TM)$, then $\omega = 0$.*

We also consider the Hodge type Laplacian related to $\nabla^P$, defined in [13] by

$$\Delta_H^P = \delta^P d^P + d^P \delta^P,$$

where

$$d^P \omega(X_0, \ldots, X_k) = \sum_i (-1)^i (\nabla_{PX_i} \omega)(X_0, \ldots, X_{i-1}, X_{i+1}, \ldots X_k),$$
$$\delta^P \omega(X_2, \ldots, X_k) = -\sum_i (\nabla_{Pe_i} \omega)(e_i, X_2, \ldots, X_k).$$

Similarly to Equations (58) and (59) in [9], we can state the following

**Lemma 3.** *For a statistical P-structure the following equalities are satisfied:*

$$
\begin{aligned}
&a) \quad d^P = \overline{d}^P = \check{d}^P, \\
&b) \quad \delta^P = \overline{\delta}^P - \iota_E = \check{\delta}^P + \iota_E, \\
&c) \quad \Delta_H^P = \overline{\Delta}_H^P + \mathcal{L}_E^P = \check{\Delta}_H^P - \mathcal{L}_E^P,
\end{aligned}
\qquad (29)
$$

*where $\mathcal{L}^P := \overline{d}^P \iota - \iota \overline{d}^P$ is the modified Lie derivative.*

**Proof.** From (12) and (26) we get equalities (29) (a). Next, we obtain

$$\overline{\delta}^P \omega = -\sum_i \overline{\nabla}_{e_i}^P \iota_{e_i} \omega = -\sum_i \nabla_{e_i}^P \iota_{e_i} \omega - \sum_i K_{e_i} \iota_{e_i} \omega = \delta^P \omega + \iota_E \omega.$$

For the second term, we have used (14). From this and $\check{\nabla}^P = \nabla^P - K$ the equalities (29) (b) follow. Finally, we calculate the following:

$$\overline{\Delta}_H^P = \overline{d}^P \check{\delta}^P + \check{\delta}^P \overline{d}^P = \overline{d}^P (\delta^P - \iota_E) + (\delta^P - \iota_E) \overline{d}^P = \Delta^P - \mathcal{L}_E^P.$$

From this and $\check{\nabla}^P = \nabla^P - K$ equalities (29) (c) follow. $\square$

The following proposition extends result for regular case, $P = \mathrm{id}_{TM}$ and $K = 0$ in [21].

**Proposition 8.** *Let $(M, g)$ be a complete non-compact Riemannian manifold endowed with a vector field $X$ such that $\overline{\mathrm{div}}_P X \geq 0$ (or $\overline{\mathrm{div}}_P X \leq 0$), where $P \in \mathrm{End}(TM)$ such that conditions (17) and $\|PX\|_g \in \mathrm{L}^1(M, g)$ hold. Then, $\overline{\mathrm{div}}_P X \equiv 0$.*

**Proof.** Let $\omega$ be the $(n-1)$-form in $M$ given by $\omega = \iota_{PX} d\,\mathrm{vol}_g$, i.e., the contraction of the volume form $d\,\mathrm{vol}_g$ in the direction of $PX$. If $\{e_1, \ldots, e_n\}$ is an orthonormal frame on an open set $U \subset M$ with coframe $\omega_1, \ldots, \omega_n$, then

$$\iota_{PX} d\,\mathrm{vol}_g = \sum_{i=1}^n (-1)^{i-1} \langle PX, e_i \rangle \, \omega_1 \wedge \ldots \wedge \omega_{i-1} \wedge \omega_{i+1} \wedge \ldots \wedge \omega_n.$$

Since the $(n-1)$-forms $\omega_1 \wedge \ldots \wedge \omega_{i-1} \wedge \omega_{i+1} \wedge \ldots \wedge \omega_n$ are orthonormal in $\Omega^{n-1}(M)$, we get $\|\omega\|_g^2 = \sum_{i=1}^n \langle PX, e_i \rangle^2 = \|PX\|_g^2$. Thus, $\|\omega\|_g \in L^1(M, g)$ and

$$d\omega = d(\iota_{PX}\, d\operatorname{vol}_g) = (\overline{\operatorname{div}}_P X)\, d\operatorname{vol}_g,$$

see (18). There exists a sequence of domains $B_i$ on $M$ such that $M = \bigcup_{i\geq 1} B_i$, $B_i \subset B_{i+1}$ and $\lim_{i\to\infty} \int_{B_i} d\omega = 0$, see [22]. Then

$$\int_{B_i} (\overline{\operatorname{div}}_P X)\, d\operatorname{vol}_g \overset{(18)}{=} \int_{B_i} \operatorname{div}(PX)\, d\operatorname{vol}_g = \int_{B_i} d\omega \to 0.$$

But since $\overline{\operatorname{div}}_P X \geq 0$ on $M$, it follows that $\overline{\operatorname{div}}_P X = 0$ on $M$. $\square$

We call $\overline{\Delta}^P f = \overline{\operatorname{div}}_P(\overline{\nabla}^P f)$ the *P-Laplacian for functions*. Using (3), we have

$$\overline{\Delta}^P f = \Delta^P f + (PE)(f), \tag{30}$$

that generalizes Lemma 6.1 in [9] for regular case, $P = \operatorname{id}_{TM}$.

Consider the following system of singular distributions on a smooth manifold $M$: $\mathcal{D}_1 = \mathcal{D}$, $\mathcal{D}_2 = \mathcal{D}_1 + [\mathcal{D}, \mathcal{D}_1]$, etc. The distribution $\mathcal{D}$ is said to be *bracket-generating* of the step $r \in \mathbb{N}$ if $\mathcal{D}_r = TM$, e.g., [2]. Note that integrable distributions, i.e., $[X, Y] \in \mathfrak{X}_{\mathcal{D}}$ $(X, Y \in \mathfrak{X}_{\mathcal{D}})$, are not bracket-generating. The condition $\overline{\nabla}^P f = 0$ means that $f \in C^2(M)$ is constant along the (integral curves of) $\mathcal{D}$; moreover, if $\mathcal{D}$ is bracket-generating then $f = \operatorname{const}$ on $M$.

The next theorem extends the well-known classical result on subharmonic functions and generalizes Theorem 1 in [13] (see also [21] for $P = \operatorname{id}_{TM}$ and $K = 0$).

**Theorem 3.** *Let conditions* (17) *hold for a statistical P-connection* $\overline{\nabla}^P$, *and let* $f \in C^2(M)$ *satisfy either* $\overline{\Delta}^P f \geq 0$ *or* $\overline{\Delta}^P f \leq 0$. *Suppose that any of the following conditions hold:*
   *(a) $(M, g)$ is closed;*
   *(b) $(M, g)$ is complete non-compact, $\|P\overline{\nabla}^P f\|$ and $\|f P\overline{\nabla}^P f\|$ belong to $L^1(M, g)$.*
   *Then, $\overline{\nabla}^P f = 0$; moreover, if $P(TM)$ is bracket-generating, then $f = \operatorname{const}$.*

**Proof.** This is as for Theorem 1 in [13]. Set $X = \overline{\nabla}^P f$, then $\overline{\Delta}^P f = \overline{\operatorname{div}}_P X$.
   (a) Using Theorem 2, we get $\overline{\Delta}^P f \equiv 0$. By the equality with $Y = \overline{\nabla}^P f$,

$$\overline{\operatorname{div}}_P(f \cdot Y) = f \cdot \overline{\operatorname{div}}_P Y + \langle \overline{\nabla}^P f,\, Y \rangle \tag{31}$$

and again Theorem 2 with $X = f\overline{\nabla}^P f$, we get $(\overline{\nabla}^P f, \overline{\nabla}^P f)_{L^2} = 0$, hence $\overline{\nabla}^P f = 0$.
   (b) By Proposition 8 with $X = \overline{\nabla}^P f$ and condition $\|P\overline{\nabla}^P f\| \in L^1(M, g)$, we get $\overline{\Delta}^P f \equiv 0$. Using (31) with $Y = \overline{\nabla}^P f$, Proposition 8 with $X = f\overline{\nabla}^P f$ and condition $\|f P\overline{\nabla}^P f\| \in L^1(M, g)$, we get $(\overline{\nabla}^P f, \overline{\nabla}^P f)_{L^2} = 0$, hence $\overline{\nabla}^P f = 0$. If the distribution $P(TM)$ is bracket-generating, then using Chow's theorem [23] completes the proof for both cases. $\square$

## 6. The Modified Curvature Tensor

**Definition 5.** *Define the second P-derivative of an $(s, k)$-tensor $S$ as the $(s, k+2)$-tensor*

$$(\overline{\nabla}^P)^2_{X,Y} S = \overline{\nabla}^P_X (\overline{\nabla}^P_Y S) - \overline{\nabla}^P_{\overline{\nabla}^P_X Y} S.$$

*Define the P-curvature tensor of $\overline{\nabla}^P$ by*

$$\begin{aligned}
\overline{R}^P_{X,Y} Z &= (\overline{\nabla}^P)^2_{X,Y} Z - (\overline{\nabla}^P)^2_{Y,X} Z \\
&= \overline{\nabla}^P_X \overline{\nabla}^P_Y Z - \overline{\nabla}^P_Y \overline{\nabla}^P_X Z - \overline{\nabla}^P_{[X,Y]_P} Z, \quad X, Y, Z \in \mathfrak{X}_M,
\end{aligned}$$

*see (49), second formula, with $\rho = P$, and set*

$$\overline{R}^{P}(X, Y, Z, W) = \langle \overline{R}^{P}_{X,Y} Z, W \rangle, \quad X, Y, Z, W \in \mathfrak{X}_M. \tag{32}$$

*The P-Ricci curvature tensor of $\overline{\nabla}^{P}$ is defined by the standard way:*

$$\overline{\mathrm{Ric}}^{P}(X) = \sum_i \overline{R}^{P}_{X,e_i} e_i, \quad \overline{\mathrm{Ric}}^{P}(X, Y) = \sum_i \overline{R}^{P}(X, e_i, e_i, Y). \tag{33}$$

The formula of the action of $\overline{R}^{P}$ on $(0, k)$-tensor fields is similar to the formula of the action of $R$ (mentioned in the Introduction),

$$(\overline{R}^{P}_{X,Y} S)(X_1, \ldots, X_k) = \overline{\mathfrak{D}}^{P}(X, Y)(S(X_1, \ldots, X_k)) - \sum_i S(X_1, \ldots \overline{R}^{P}_{X,Y} X_i, \ldots, X_k). \tag{34}$$

To simplify the calculations, in the rest of the article we assume that the tensor $K$ satisfies the following Codazzi type condition:

$$(\nabla_{PX} K)_Y Z = (\nabla_{PY} K)_X Z, \quad X, Y, Z \in \mathfrak{X}_M. \tag{35}$$

Here, $(\nabla_{PX} K)_Y Z = \nabla_{PX}(K_Y Z) - K_{\nabla_{PX}Y} Z - K_Y(\nabla_{PX} Z)$. Note that $[K_X, K_Y] : TM \to TM$ is a skew-symmetric endomorphism for a statistical $P$-structure.

The following result generalizes Proposition 6 in [16].

**Proposition 9.** *For a statistical P-structure, we have*

1. $\overline{R}^{P}_{X,Y} Z = R_{PX,PY} Z + [K_X, K_Y](Z); \quad (\overline{R}^{P}_{X,Y} \omega)(Z) = -\omega(\overline{R}^{P}_{X,Y} Z - [K_X, K_Y](Z));$

   *hence,* $\langle \overline{R}^{P}_{X,Y} Z, W \rangle = -\langle \overline{R}^{P}_{X,Y} W, Z \rangle,$
2. $\overline{R}^{P}_{X,Y} f = 0$ *for any* $f \in C^2(M); \quad \overline{R}^{P}_{X,Y} g = 0;$
3. *for every $(1, k)$-tensor S we have*

$$(\overline{R}^{P}_{X,Y} S)(Z_1, \ldots, Z_k) = (R_{PX,PY} S)(Z_1, \ldots, Z_k)$$
$$+ [K_X, K_Y](S(Z_1, \ldots, Z_k)) - \sum_i S(Z_1, \ldots [K_X, K_Y](Z_i), \ldots, Z_k).$$

4. $\overline{R}^{P}(X, Y, Z, W) = R(PX, PY, Z, W) + \langle [K_X, K_Y](Z), W \rangle;$
5. $\overline{R}^{P}(X, Y, Z, W) = -\overline{R}^{P}(Y, X, Z, W) = -\overline{R}^{P}(X, Y, W, Z)$, *where* $X, Y, Z, W \in \mathfrak{X}_M, \omega \in \Lambda^1(TM)$ *and* $f \in C^2(M).$

**Proof.** 1. Since $P\overline{[X, Y]}_P = [PX, PY]$, see definition of $\overline{\mathfrak{D}}^{P}$, we have

$$\overline{R}^{P}_{X,Y} Z = R_{PX,PY} Z + (\nabla_{PX} K)_Y Z - (\nabla_{PY} K)_X Z + [K_X, K_Y](Z),$$
$$(\overline{R}^{P}_{X,Y} \omega)(Z) = -\omega(R_{PX,PY} Z) + \omega((\nabla_{PX} K)_Y Z - (\nabla_{PY} K)_X Z + [K_X, K_Y](Z)).$$

From this and (35) the first claim follows. Since $[K_X, K_Y] : TM \to TM$ is skew-symmetric, then $\overline{R}^{P}_{X,Y}$ is also skew-symmetric.

2. We calculate

$$\overline{R}^{P}_{X,Y} f = PX(PY(f)) - PY(PX(f)) - (P\overline{[X, Y]}_P)f = \overline{\mathfrak{D}}^{P}(X, Y)f = 0.$$

Next, using 1. we obtain

$$\langle \overline{R}^{P}_{X,Y} Z, W \rangle = \langle R_{PX,PY} Z, W \rangle + \langle [K_X, K_Y](Z), W \rangle.$$

Similarly, $\langle \overline{R}^{\,P}_{X,Y}\, W,\, Z \rangle = \langle R_{PX,PY}\, W,\, Z \rangle + \langle [K_X, K_Y](W),\, Z \rangle$ . By this and (34), we get

$$
\begin{aligned}
(\overline{R}^{\,P}_{X,Y}\, g)(Z, W) &= -\langle \overline{R}^{\,P}_{X,Y}\, Z, W \rangle - \langle Z, \overline{R}^{\,P}_{X,Y}\, W \rangle \\
&= (R_{PX,PY}\, g)(Z, W) - \langle [K_X, K_Y](Z), W \rangle - \langle [K_X, K_Y](W), Z \rangle \\
&= (R_{PX,PY}\, g)(Z, W).
\end{aligned}
$$

Using $R_{PX,PY}\, g = 0$ and the property (10), we obtain $\overline{R}^{\,P}_{X,Y}\, g = 0$.
3. From the above and (34) the claim follows.
4. The equality follows from (32) and 1.
5. Since $\overline{R}^{\,P}_{X,Y}\, Z = -\overline{R}^{\,P}_{Y,X}\, Z$, see 1., the first equality follows. For the second one, we use 2:

$$
0 = (\overline{R}^{\,P}_{X,Y}\, g)(Z, Z) = -2\langle \overline{R}^{\,P}_{X,Y} Z,\, Z \rangle;
$$

thus, the claim follows from the equality $\langle \overline{R}^{\,P}_{X,Y}(Z + W),\, Z + W \rangle = 0$.   $\square$

Similarly, we define the *P*-curvature tensor of the conjugate *P*-connection $\check{\nabla}^{\,P}$,

$$
\check{R}^{\,P}_{X,Y}\, Z = \check{\nabla}^{\,P}_X \check{\nabla}^{\,P}_Y\, Z - \check{\nabla}^{\,P}_Y \check{\nabla}^{\,P}_X\, Z - \check{\nabla}^{\,P}_{\overset{\smile}{[X,Y]}_P}\, Z, \quad X, Y, Z \in \mathfrak{X}_M.
$$

The following curvature type tensor (depending on *P* only) has been introduced in [13]:

$$
R^{\,P}_{X,Y}\, Z = \nabla_{PX}\nabla_{PY}\, Z - \nabla_{PY}\nabla_{PX}\, Z - \nabla_{P[X,Y]_P}\, Z, \quad X, Y, Z \in \mathfrak{X}_M,
$$

Since we assume $\overline{\mathfrak{D}}^{\,P} = 0$ then $R^{\,P}_{X,Y} = R_{PX,PY}$ holds. By the above,

$$
\overline{R}^{\,P}_{X,Y} = R^{\,P}_{X,Y} + [K_X, K_Y], \quad \check{R}^{\,P}_{X,Y} = R^{\,P}_{X,Y} - [K_X, K_Y].
$$

Thus,

$$
\overline{R}^{\,P}_{X,Y} + \check{R}^{\,P}_{X,Y} = 2\,R^{\,P}_{X,Y}, \qquad \langle \overline{R}^{\,P}_{X,Y}\, Z, W \rangle = -\langle \check{R}^{\,P}_{X,Y}\, W, Z \rangle,
$$

and $\overset{\smile}{\mathrm{Ric}}^{\,P}(X, Y) = \overline{\mathrm{Ric}}^{\,P}(X, Y)$ when (35) holds. The Ricci tensor of $\nabla^P$ was defined in [13] by

$$
\mathrm{Ric}^P(X, Y) = \sum_i R(PX, Pe_i, e_i, Y).
$$

**Proposition 10.** *For a statistical P-structure, we have*

$$
\overline{\mathrm{Ric}}^{\,P}(X, Y) = \mathrm{Ric}^P(X, Y) + \langle K_X Y, E \rangle - \langle K_X, K_Y \rangle. \tag{36}
$$

*Thus, $\overline{\mathrm{Ric}}^{\,P}$ is symmetric if and only if $\mathrm{Ric}^P$ is symmetric.*

**Proof.** Using symmetry of *K*, we have

$$
\begin{aligned}
\overline{\mathrm{Ric}}^{\,P}(X, Y) &= \sum_i \overline{R}^{\,P}(X, e_i, e_i, Y) = \sum_i \left( R(PX, Pe_i, e_i, Y) + \langle [K_X, K_{e_i}](e_i), Y \rangle \right) \\
&= \mathrm{Ric}^P(X, Y) + \sum_i \langle [K_X, K_{e_i}](e_i), Y \rangle = \mathrm{Ric}^P(X, Y) + \langle K_X Y, E \rangle - \langle K_X, K_Y \rangle.
\end{aligned}
$$

From the above the claim follows.   $\square$

The endomorphism *P* of *TM* induces endomorphisms $\mathcal{P}$ and its adjoint $\mathcal{P}^*$ of $\Lambda^2(TM)$:

$$
\mathcal{P}(X \wedge Y) = PX \wedge PY, \quad \mathcal{P}^*(X \wedge Y) = P^*X \wedge P^*Y,
$$

see [13]. The curvature tensor $R_{X,Y}$ can be seen as a self-adjoint linear operator $\mathcal{R}$ on the space $\Lambda^2(TM)$ of bivectors, called the *curvature operator*, e.g., [7,19]. Similarly, we consider $\overline{R}^{\,P}_{X,Y} = R_{PX,PY} + [K_X, K_Y]$

as a linear operator or as a corresponding bilinear form on $\Lambda^2(TM)$. For this, using skew-symmetry of $[K_X, K_Y]$ for a statistical $P$-connection, define a linear operator $\mathcal{K}$ on $\Lambda^2(TM)$ by

$$\langle \mathcal{K}(X \wedge Y), Z \wedge W \rangle = \langle [K_X, K_Y](Z), W \rangle,$$

and observe $\mathcal{K}^* = \mathcal{K}$ (symmetry). Put $\overline{\mathcal{R}}^{\,P} = \mathcal{R}\,\mathcal{P} + \mathcal{K}$ and $\check{\mathcal{R}}^P = \mathcal{R}\,\mathcal{P} - \mathcal{K}$, i.e.,

$$\overline{\mathcal{R}}^{\,P}(X \wedge Y) = \mathcal{R}\,\mathcal{P}(X \wedge Y) + \mathcal{K}(X \wedge Y) = \mathcal{R}(PX \wedge PY) + \mathcal{K}(X \wedge Y),$$
$$\overline{\mathcal{R}}^{\,P}(X \wedge Y, Z \wedge W) = \langle \overline{\mathcal{R}}^{\,P}(X \wedge Y), Z \wedge W \rangle,$$
$$\check{\mathcal{R}}^P(X \wedge Y) = \mathcal{R}\,\mathcal{P}(X \wedge Y) - \mathcal{K}(X \wedge Y) = \mathcal{R}(PX \wedge PY) - \mathcal{K}(X \wedge Y),$$
$$\check{\mathcal{R}}^P(X \wedge Y, Z \wedge W) = \langle \check{\mathcal{R}}^{\,P}(X \wedge Y), Z \wedge W \rangle.$$

The above $\overline{\mathcal{R}}^{\,P}$ generalizes $\mathcal{R}^{\,P} = \mathcal{R}\,\mathcal{P}$, having the properties, see [13],

$$\mathcal{R}^{\,P}(X \wedge Y) = \mathcal{R}\,\mathcal{P}(X \wedge Y) = \mathcal{R}(PX \wedge PY),$$
$$\mathcal{R}^{\,P}(X \wedge Y, Z \wedge W) = \langle \mathcal{R}^{\,P}(X \wedge Y), Z \wedge W \rangle,$$
$$\langle \mathcal{R}^{\,P}(X \wedge Y), Z \wedge W \rangle = \langle \mathcal{R}(PX \wedge PY), Z \wedge W \rangle = R^P(X, Y, W, Z).$$

Using known properties of $\mathcal{R}$ and property 4. of $\overline{R}^{\,P}$, we have

$$\langle \overline{\mathcal{R}}^{\,P}(X \wedge Y), Z \wedge W \rangle = \langle \mathcal{R}(PX \wedge PY) + \mathcal{K}(X \wedge Y), Z \wedge W \rangle = \overline{R}^{\,P}(X, Y, W, Z),$$
$$\langle \check{\mathcal{R}}^P(X \wedge Y), Z \wedge W \rangle = \langle \mathcal{R}(PX \wedge PY) - \mathcal{K}(X \wedge Y), Z \wedge W \rangle = \check{R}^P(X, Y, W, Z).$$

Note that if $\mathcal{P}^* \mathcal{R} \neq \mathcal{R}\,\mathcal{P}$ then $\overline{\mathcal{R}}^{\,P}$ on $\Lambda^2(TM)$ is not self-adjoint:

$$(\overline{\mathcal{R}}^{\,P})^* = (\mathcal{R}\,\mathcal{P} + \mathcal{K})^* = \mathcal{P}^* \mathcal{R} + \mathcal{K} \neq \mathcal{R}\,\mathcal{P} + \mathcal{K} = \overline{\mathcal{R}}^{\,P}.$$

## 7. The Weitzenböck Type Curvature Operator

Here, we use the $P$-connection $\overline{\nabla}^{\,P}$ to introduce the central concept of the paper: the Weitzenböck type curvature operator on tensors. We generalize the Weitzenböck curvature operator (2), (see also [9] for statistical manifolds when $P = \mathrm{id}_{TM}$, and [13] for distributions when $K = 0$) for the case of distributions with statistical structure.

**Definition 6.** *Define the $P$-Weitzenböck curvature operator on $(0,k)$-tensors $S$ over $(M,g)$ by*

$$\overline{\Re}^{\,P}(S)(X_1, \ldots, X_k) = \sum_{a=1}^k \sum_i (\overline{R}^{\,P}_{e_i, X_a} S)(X_1, \ldots, \underbrace{e_i}_{a}, \ldots, X_k). \tag{37}$$

*The operators $\check{\Re}^P$ and $\Re^P$ are defined similarly using $P$-connections $\check{\nabla}^{\,P}$ and $\nabla^P$.*

For a differential form $\omega$, the $\overline{\Re}^{\,P}(\omega)$ is skew-symmetric. Note that $\overline{\Re}^{\,P}$ reduces to $\overline{\mathrm{Ric}}^{\,P}$ when evaluated on $(0,1)$-tensors, i.e., $k = 1$. For $k \geq 2$ using (34), the formula from (37) reads as

$$\overline{\Re}^{\,P}(S)(X_1, \ldots, X_k) = -2 \sum_{i,j,a;b<a} \overline{R}^{\,P}(e_i, X_a, e_j, X_b) \cdot S(X_1, \ldots, \underbrace{e_j}_{b}, \underbrace{\ldots, e_i}_{a-b}, \ldots, X_k)$$

$$+ \sum_{i,a} \overline{\mathrm{Ric}}^{\,P}(e_i, X_a) \cdot S(X_1, \ldots, \underbrace{e_i}_{a}, \ldots, X_k), \tag{38}$$

or, in coordinates, $\overline{\Re}^{\,P}(S)_{i_1,\ldots,i_k} = -2 \sum_{a<b} \overline{R}^{\,P}_{j\,i_a\,p\,i_b} S_{i_1\ldots\,{}^{j}\ldots\,{}^{p}\ldots\,i_k} + \sum_a \overline{\mathrm{Ric}}^{\,P}_{i_a j} S_{i_1\ldots\,{}^{j}\ldots\,i_k}$.

The following lemma represents $\overline{\Re}^{\,P}$ using $\Re^P$ and $K$.

**Lemma 4.** *For a statistical P-structure, let* (21) *hold. Then we have*

$$\overline{\mathfrak{R}}^P = \mathfrak{R}^P - \mathfrak{K}, \tag{39}$$

*where the operator $\mathfrak{K}$ acts on k-forms $\omega$ over $(M, g)$ by*

$$(\mathfrak{K}\omega)(X_1, \ldots, X_k) = \sum_{a=1}^{k} \sum_j \langle K_{X_a}, K_{e_j} \rangle \, \omega(\underbrace{X_1, \ldots, e_j}_{a}, \ldots, X_k)$$

$$+ 2 \sum_{i,j,b<a} \left( \langle K_{X_a} e_j, K_{X_b} e_i \rangle - \langle K_{e_i} e_j, K_{X_a} X_b \rangle \right) \omega(\underbrace{X_1, \ldots, e_j}_{b}, \underbrace{\ldots, e_i}_{a-b} \ldots, X_k), \tag{40}$$

*when $k \geq 2$, and $(\mathfrak{K}\omega)(X) = \sum_j \langle K_X, K_{e_j} \rangle \, \omega(e_j)$ when $k = 1$.*

**Proof.** Using 1. of Proposition 9 and (36), we have

$$\overline{R}^P(e_i, X_a, e_j, X_b) = R^P(e_i, X_a, e_j, X_b) + \langle [K_{e_i}, K_{X_a}](e_j), X_b \rangle,$$
$$\overline{\mathrm{Ric}}^P(e_i, X_a) = \mathrm{Ric}^P(e_i, X_a) + \langle K_{e_i} X_a, E \rangle - \langle K_{e_i}, K_{X_a} \rangle.$$

Substituting the above equalities in (37) (and using linearity in the curvature) yields (39) with

$$(\mathfrak{K}\omega)(X_1, \ldots, X_k) = \sum_{i,a} \left( \langle K_{X_a}, K_{e_j} \rangle - \langle K_{X_a} e_j, E \rangle \right) \omega(\underbrace{X_1, \ldots, e_i}_{a}, \ldots, X_k)$$

$$+ 2 \sum_{i,j; b<a} \left( \langle K_{X_a} e_j, K_{X_b} e_i \rangle - \langle K_{e_i} e_j, K_{X_a} X_b \rangle \right) \omega(\underbrace{X_1, \ldots, e_j}_{b}, \underbrace{\ldots, e_i}_{a-b} \ldots, X_k),$$

that is (40) when $E = 0$.  □

The following theorem generalizes (1) to the case of distributions and Theorem 2 in [13] to the case of statistical *P*-structure.

**Theorem 4.** *For a statistical P-structure, let* (21) *hold. Then, the following Weitzenböck type decomposition formula is valid for any k-form $\omega$:*

$$\overline{\Delta}_H^P \omega = \check{\nabla}^{*P} \overline{\nabla}^P \omega + \overline{\mathfrak{R}}^P(\omega). \tag{41}$$

**Proof.** Similarly to the proof of Theorem 9.4.1 in [19] for $\omega \in \Lambda^k(TM)$, or Theorem 2 in [13], we find

$$\overline{d}^P \check{\delta}^P \omega(X_1, \ldots, X_k) = -\sum_j \overline{d}^P \check{\nabla}_{e_j}^P \omega(e_j, X_1, \ldots, X_k)$$

$$= -\sum_j \overline{d}^P (\overline{\nabla}_{e_j}^P - 2K_j) \omega(e_j, X_1, \ldots, X_k)$$

$$= -\sum_j \overline{d}^P \overline{\nabla}_{e_j}^P \omega(e_j, X_1, \ldots, X_k) - 2(\overline{d}^P \iota_E \omega)(X_1, \ldots, X_k)$$

$$= \sum_j \sum_{a=0}^{k-1} (-1)^a \overline{\nabla}_{X_{a+1}}^P \overline{\nabla}_{e_j}^P \omega(e_j, X_1, \ldots X_a, X_{a+2} \ldots, X_k) - 2(\overline{d}^P \iota_E \omega)(X_1, \ldots, X_k)$$

$$= -\sum_j \sum_{a=0}^{k-1} \overline{\nabla}_{X_{a+1}}^P \overline{\nabla}_{e_j}^P \omega(\underbrace{X_1, \ldots e_j}_{a+1}, \ldots, X_k) - 2(\overline{d}^P \iota_E \omega)(X_1, \ldots, X_k)$$

$$= -\sum_{j,a} ((\overline{\nabla}^P)_{X_{a+1}, e_j}^2 \omega)(\underbrace{X_1, \ldots e_j}_{a+1}, \ldots, X_k) - 2(\overline{d}^P \iota_E \omega)(X_1, \ldots, X_k),$$

where $E = \sum_i K_{e_i} e_i$ (see Section 3), and

$$
\begin{aligned}
\check{\delta}^P \bar{d}^P \omega(X_1,\ldots,X_k) &= \check{\nabla}^{*P} \bar{d}^P \omega(X_1,\ldots,X_k) = (\overline{\nabla}^{*P} - 2\,\iota_E)\bar{d}^P \omega(X_1,\ldots,X_k) \\
&= \overline{\nabla}^{*P}(\bar{d}^P \omega)(X_1,\ldots,X_k) - 2\,\iota_E \bar{d}^P \omega(X_1,\ldots,X_k) \\
&= -\sum_j \overline{\nabla}^P_{e_j}(\bar{d}^P \omega)(e_j, X_1,\ldots,X_k) - 2\,\iota_E(\bar{d}^P \omega)(X_1,\ldots,X_k) \\
&= -\sum_j \overline{\nabla}^P_{e_j} \overline{\nabla}^P_{e_j} \omega(X_1,\ldots,X_k) \\
&\quad + \sum_j \sum_{a=0}^{k-1}(-1)^a \overline{\nabla}^P_{e_j} \overline{\nabla}^P_{X_{a+1}} \omega(e_j, X_1,\ldots,X_a, X_{a+2},\ldots,X_k) - 2\,\iota_E(\bar{d}^P \omega)(X_1,\ldots,X_k) \\
&= (\overline{\nabla}^{*P} \overline{\nabla}^P \omega)(X_1,\ldots,X_k) + \sum_{j,a}((\overline{\nabla}^P)^2_{e_j, X_{a+1}} \omega)(X_1,\ldots,X_k) - 2\,\iota_E(\bar{d}^P \omega)(X_1,\ldots,X_k).
\end{aligned}
$$

Thus, if (17) is assumed, then using $\overline{\nabla}^{*P} \overline{\nabla}^P = (\check{\nabla}^{*P} + 2\,\iota_E)\overline{\nabla}^P = \check{\nabla}^{*P} \overline{\nabla}^P + 2\overline{\nabla}^P_E$, we have

$$
\overline{\Delta}^P_H \omega = \check{\nabla}^{*P} \overline{\nabla}^P \omega + \overline{\mathfrak{R}}^P \omega - 2\,\mathcal{L}_E \omega + 2\overline{\nabla}^P_E \omega. \tag{42}
$$

Using assumption $E = 0$, we reduce (42) to a shorter form (41). $\quad\square$

Next, we extend the well-known Bochner–Weitzenböck formula (and generalize Proposition 7 in [13] where $K = 0$) to the case of distributions with a statistical $P$-structure.

**Proposition 11.** *For a statistical $P$-structure, let (21) hold. Then the following modified Bochner–Weitzenböck formula for k-forms is valid:*

$$
\frac{1}{2}\overline{\Delta}^P(\|\omega\|^2) = -\langle \overline{\Delta}^P_H \omega, \omega \rangle + \langle \overline{\mathfrak{R}}^P(\omega), \omega \rangle + \|(\overline{\nabla}^P - K)\,\omega\|^2 + \langle \mathfrak{K}\omega, \omega \rangle. \tag{43}
$$

**Proof.** Applying Proposition 7 in [13], (29) (c) and (30), we find

$$
\begin{aligned}
\frac{1}{2}\overline{\Delta}^P(\|\omega\|^2) - (PE)(\|\omega\|^2) &= \frac{1}{2}\Delta^P(\|\omega\|^2) \\
&= -\langle \Delta^P_H \omega, \omega \rangle + \langle \mathfrak{R}^P(\omega), \omega \rangle + \|\nabla^P \omega\|^2 \\
&= -\langle (\overline{\Delta}^P_H + \mathcal{L}^P_E)\,\omega, \omega \rangle + \langle (\overline{\mathfrak{R}}^P + \mathfrak{K})\,\omega, \omega \rangle + \|(\overline{\nabla}^P - K)\,\omega\|^2.
\end{aligned}
$$

Using assumption $E = 0$, we reduce the above to a shorter form (43). $\quad\square$

**Remark 3.** (a) For $k = 1$, we have $(\mathfrak{K}\omega)(X) = \sum_i \langle K_X, K_{e_i} \rangle \omega(e_i)$. Thus,

$$
\langle \mathfrak{K}\omega,\,\omega \rangle = \sum_{i,j}\langle K_{e_i}, K_{e_j} \rangle \omega(e_i)\,\omega(e_j) = \|K_{\omega^\sharp}\|^2 \geq 0,
$$

where $\omega^\sharp = \sum_i \omega(e_i)e_i$ for any $\omega \in \Lambda^1(M)$.

(b) If $\omega$ is a $P$-harmonic $k$-form on a closed manifold $M$ and $\langle(\overline{\mathfrak{R}}^P + \mathfrak{K})(\omega), \omega \rangle \geq 0$, then $\overline{\Delta}^P(\|\omega\|^2) = 0$, $(\overline{\nabla}^P - K)\,\omega = 0$ and $(\overline{\mathfrak{R}}^P + \mathfrak{K})\,\omega = 0$, see (43). By Theorem 3, $\overline{\nabla}^P\|\omega\| = 0$; moreover, if $P(TM)$ is bracket-generating, then $\|\omega\| = $ const on $M$.

**Example 4.** *For vector fields and 1-forms, $\overline{\mathfrak{R}}^P$ reduces to the kind of usual Ricci curvature, see (33) and (38). We have $\overline{\mathfrak{R}}^P(\omega)(X) = \omega(\overline{\mathrm{Ric}}^P(X))$ for any $\omega \in \Lambda^1(M)$; thus, (41) reads as*

$$
\overline{\Delta}^P_H \omega = \check{\nabla}^{*P} \overline{\nabla}^P \omega + \overline{\mathrm{Ric}}^P(\omega^\sharp).
$$

For every bivector $X \wedge Y \in \Lambda^2(TM)$, we build a map $\overline{\mathcal{R}}^P(X \wedge Y) : \mathfrak{X}_M \to \mathfrak{X}_M$, given by

$$
\begin{aligned}
\langle \overline{\mathcal{R}}^P(X \wedge Y)Z, W \rangle &= \langle \overline{\mathcal{R}}^P(X \wedge Y), W \wedge Z \rangle = \overline{R}^P(X, Y, Z, W) \\
&= R(PX, PY, Z, W) + \langle [K_X, K_Y](Z), W \rangle.
\end{aligned}
$$

Since bivectors are generators of the vector space $\Lambda^2(TM)$, we obtain in this way a map $\overline{\mathcal{R}}^P(\xi)$ : $\mathfrak{X}_M \to \mathfrak{X}_M$ (similarly to algebraic curvature operator $\mathcal{R}(\xi)$).

The following lemma generalizes Lemma 3 in [13].

**Lemma 5.** *The map* $\overline{\mathcal{R}}^P(\xi)$, *where* $\xi \in \Lambda^2(TM)$, *is skew-symmetric:*

$$\langle \overline{\mathcal{R}}^P(\xi)W, Z \rangle = -\langle \overline{\mathcal{R}}^P(\xi)Z, W \rangle.$$

**Proof.** It suffices to check the statement for the generators. We have, using Proposition 9,

$$\langle \overline{\mathcal{R}}^P(X \wedge Y)Z, W \rangle = R(PX, PY, Z, W) + \langle [K_X, K_Y](Z), W \rangle$$
$$= -R(PX, PY, W, Z) - \langle [K_X, K_Y](W), Z \rangle = -\langle \overline{\mathcal{R}}^P(X \wedge Y)W, Z \rangle.$$

Thus, the statement follows. □

The associated *P-curvature operator* is given by

$$\langle \overline{\mathcal{R}}^P(X \wedge Y), Z \wedge W \rangle = R(PX, PY, W, Z) - \langle [K_X, K_Y](Z), W \rangle.$$

To simplify calculations, we assume that $\mathfrak{so}(TM)$ is endowed with metric induced from $\Lambda^2(TM)$, e.g., [13]. If $L \in \mathfrak{so}(TM)$, then

$$(L\,S)(X_1, \dots, X_k) = -\sum\nolimits_i S(X_1, \dots, L(X_i), \dots, X_k). \tag{44}$$

Let $\{\xi_a\}$ be an orthonormal base of skew-symmetric transformations such that $(\xi_a)_x \in \mathfrak{so}(T_xM)$ for $x$ in an open set $U \subset M$. By (44), for any $(0, k)$-tensor $S$,

$$(\xi_\alpha S)(X_1, \dots, X_k) = -\sum\nolimits_i S(X_1, \dots, \xi_\alpha(X_i), \dots, X_k);$$

The $\overline{\mathcal{R}}^P(X \wedge Y)$ on $\Lambda^2(TM)$ can be decomposed using $\{\xi_a\}$.

**Lemma 6** (see Lemma 4 in [13] where $K = 0$). *We have*

$$\overline{\mathcal{R}}^P(X \wedge Y) = -\sum_\alpha \left( \langle \mathcal{P}^* \mathcal{R}(\xi_\alpha)X, Y \rangle + \langle \mathcal{K}(X \wedge Y), \xi_\alpha \rangle \right)\xi_\alpha$$
$$= -\sum_\alpha \left( \langle \mathcal{R}(\xi_\alpha)PX, PY \rangle + \langle \mathcal{K}(X \wedge Y), \xi_\alpha \rangle \right)\xi_\alpha.$$

**Proof.** Using $(\overline{\mathcal{R}}^P)^* = \mathcal{P}^* \mathcal{R}$ and Lemma 5, we have:

$$\overline{\mathcal{R}}^P(X \wedge Y) = \sum_\alpha \langle \overline{\mathcal{R}}^P(X \wedge Y), \xi_\alpha \rangle \xi_\alpha$$
$$= \sum_\alpha \left( \langle \mathcal{P}^* \mathcal{R}(\xi_\alpha), X \wedge Y \rangle + \langle \mathcal{K}(X \wedge Y), \xi_\alpha \rangle \right)\xi_\alpha$$
$$= -\sum_\alpha \left( \langle \mathcal{R}(\xi_\alpha)PX, PY \rangle + \langle \mathcal{K}(X \wedge Y), \xi_\alpha \rangle \right)\xi_\alpha. \quad \square$$

Lemma 6 allows us to rewrite the operator (37). The following result generalizes Proposition 8 in [13].

**Proposition 12.** *If S is a $(0, k)$-tensor on $(M, g)$, then*

$$\overline{\mathfrak{R}}^P(S) = -\sum_\alpha \overline{\mathcal{R}}^P(\xi_a)(\xi_a S), \qquad (\overline{\mathfrak{R}}^P(S))^* = \overline{\mathfrak{R}}^{P^*}(S).$$

*In particular, if P is self-adjoint, then* $\overline{\mathfrak{R}}^P$ *is self-adjoint too.*

**Proof.** We follow similar arguments as in the proof of Lemma 9.3.3 in [19]:

$$\overline{\mathfrak{R}}^P(S)(X_1,\ldots,X_k) = \sum_{i,j}(\overline{\mathcal{R}}^P(e_j \wedge X_i)S)(\underbrace{X_1,\ldots,e_j,\ldots,X_k}_{i})$$

$$= -\sum_{i,j,\alpha}\left(\langle \mathcal{P}^*\,\mathcal{R}(\xi_\alpha)e_j, X_i\rangle + \langle \mathcal{K}(e_j \wedge X_i),\xi_\alpha\rangle\right)(\xi_\alpha S)(X_1,\ldots,e_j,\ldots,X_k)$$

$$= -\sum_{i,j,\alpha}(\xi_\alpha S)(X_1,\ldots,(\langle \mathcal{P}^*\,\mathcal{R}(\xi_\alpha)e_j, X_i\rangle e_j + \langle \mathcal{K}(e_j \wedge X_i),\xi_\alpha\rangle),\ldots,X_k)$$

$$= -\sum_{i,j,\alpha}(\xi_\alpha S)(X_1,\ldots,\langle e_j,\overline{\mathcal{R}}^P(\xi_\alpha)X_i\rangle e_j,\ldots,X_k)$$

$$= -\sum_{i,\alpha}(\xi_\alpha S)(X_1,\ldots,\overline{\mathcal{R}}^P(\xi_\alpha)X_i,\ldots,X_k) = -\sum_\alpha(\overline{\mathcal{R}}^P(\xi_\alpha)(\xi_\alpha S))(X_1,\ldots,X_k).$$

Thus, the first claim follows. Since $\mathcal{R} : \Lambda^2(TM) \to \Lambda^2(TM)$ is self-adjoint, there is a local orthonormal base $\{\xi_a\}$ of $\Lambda^2(TM)$ such that $\mathcal{R}(\xi_a) = \lambda_a\,\xi_a$. Using this base, for any $(0,k)$-tensors $S_1$ and $S_2$, we get

$$\begin{aligned}
\langle \overline{\mathfrak{R}}^P(S_2),\, S_1\rangle &= -\sum_\alpha\langle \overline{\mathcal{R}}^P(\xi_\alpha)(\xi_\alpha S_2), S_1\rangle = -\sum_\alpha\langle \xi_\alpha S_2, (\overline{\mathcal{R}}^P)^*(\xi_\alpha)S_1\rangle \\
&= \sum_\alpha\langle \xi_\alpha S_2, (\mathcal{P}^*\,\mathcal{R} + \mathcal{K})(\xi_\alpha)(S_1)\rangle \\
&= \sum_\alpha \lambda_\alpha\langle \mathcal{P}(\xi_\alpha S_2), \xi_\alpha S_1\rangle + \sum_\alpha\langle \mathcal{K}(\xi_\alpha S_2), \xi_\alpha S_1\rangle,
\end{aligned} \tag{45}$$

and, similarly, again using $\mathcal{K}^* = \mathcal{K}$,

$$\begin{aligned}
\langle S_2, \overline{\mathfrak{R}}^{P^*}(S_1)\rangle &= \sum_\alpha \lambda_\alpha\langle \xi_\alpha S_2, \mathcal{P}^*(\xi_\alpha S_1)\rangle + \sum_\alpha\langle \xi_\alpha S_2, \mathcal{K}(\xi_\alpha S_1)\rangle \\
&= \sum_\alpha \lambda_\alpha\langle \mathcal{P}(\xi_\alpha S_2), \xi_\alpha S_1\rangle + \sum_\alpha\langle \mathcal{K}(\xi_\alpha S_2), \xi_\alpha S_1\rangle.
\end{aligned}$$

Thus, the second claim follows. $\square$

The following result generalizes Corollary 9.3.4 in [19] and Proposition 10 in [13].

**Proposition 13.** *Let $(g,\overline{\nabla}^P)$ be a statistical P-structure on a manifold M.*
*(a) If $\langle \overline{\mathcal{R}}^P(S),S\rangle \geq 0$ for any $(0,k)$-tensor S, then $\langle \overline{\mathfrak{R}}^P(S),S\rangle \geq 0$.*
*(b) Moreover, if $\langle \overline{\mathcal{R}}^P(S),S\rangle \geq -\varepsilon\,\|S\|^2$ for any $(0,k)$-tensor S, where $\varepsilon > 0$, then*

$$\langle \overline{\mathfrak{R}}^P(S),S\rangle \geq -\varepsilon\,C\,\|S\|^2,$$

*where a constant C depends only on the type of S.*

**Proof.** Using (45) and a local orthonormal base $\{\xi_\alpha\}$ of $\Lambda^2(TM)$ such that $\mathcal{R}(\xi_\alpha) = \lambda_\alpha\xi_\alpha$, we get

$$\begin{aligned}
\langle \overline{\mathfrak{R}}^P(S),S\rangle &= \sum_\alpha \lambda_\alpha\langle \mathcal{P}(\xi_\alpha S), \xi_\alpha S\rangle + \sum_\alpha\langle \mathcal{K}(\xi_\alpha S), \xi_\alpha S\rangle \\
&= \sum_\alpha\langle \mathcal{P}(\xi_\alpha S), \mathcal{R}(\xi_\alpha S)\rangle + \sum_\alpha\langle \mathcal{K}(\xi_\alpha S), \xi_\alpha S\rangle \\
&= \sum_\alpha\langle \overline{\mathcal{R}}^P(\xi_\alpha S), \xi_\alpha S\rangle.
\end{aligned}$$

By conditions, $\langle \overline{\mathcal{R}}^P(\xi_\alpha S), \xi_\alpha S\rangle \geq 0$ for all $\alpha$, thus, $\langle \overline{\mathfrak{R}}^P(S),S\rangle \geq 0$, and the first claim follows. There is a constant $C > 0$ depending only on the type of the tensor and $\dim M$ such that $C\|S\|^2 \geq \sum_\alpha \|\xi_\alpha S\|^2$, see Corollary 9.3.4 in [19]. By conditions, $\langle \overline{\mathcal{R}}^P(\xi_\alpha S), \xi_\alpha S\rangle \geq -\varepsilon\,\|\xi_\alpha S\|^2$ for all $\alpha$. The above yields $\langle \overline{\mathcal{R}}^P(\xi_\alpha S), \xi_\alpha S\rangle \geq -\varepsilon\,C\,\|S\|^2$ – thus, the second claim. $\square$

The following result extends Corollary 1 in [13].

**Theorem 5.** *Let (21) be satisfied for a statistical P-structure on a closed manifold M and $\langle \overline{\mathcal{R}}^P(\omega),\omega\rangle \geq 0$ for any k-form $\omega$. Then any P-harmonic k-form on M is $\overline{\nabla}^P$-parallel.*

**Proof.** By conditions and Proposition 13(a), $\langle \overline{\mathfrak{R}}^P(\omega), \omega \rangle \geq 0$. By (41), since $\overline{\Delta}_H^P \omega = 0$, we get $\langle \breve{\nabla}^{*P} \overline{\nabla}^P \omega, \omega \rangle \leq 0$. By Proposition 5, we have $\overline{\nabla}^P \omega = 0$. $\quad\square$

The following result extends Theorem 3 with $\nabla^P$ and $k = 1$ in [13].

**Theorem 6.** *Let* (21) *be satisfied for a statistical P-connection on a complete non-compact* $(M, g)$ *and* $\|K_X\| \geq \varepsilon \|X\|$ *for some* $\varepsilon > 0$ *and all* $X \in TM$. *Suppose that* $\langle \mathcal{R}^P(\omega), \omega \rangle \geq -(\varepsilon/C) \|\omega\|^2$ *for any* 1*-form* $\omega$, *where* $C$ *is defined in Proposition* 13*(b). If* $\|P\overline{\nabla}^P(\|\omega\|^2)\| \in L^1(M, g)$ *for a P-harmonic* 1*-form* $\omega$, *then* $\nabla^P \omega = 0$.

**Proof.** By conditions, Remark 3 and Proposition 13(b),

$$\langle \mathfrak{R}^P(\omega), \omega \rangle = \langle \overline{\mathfrak{R}}^P(\omega), \omega \rangle + \langle \mathfrak{K}(\omega), \omega \rangle \geq -\varepsilon \|\omega\|^2 + \|K_{\omega^\sharp}\|^2 \geq 0.$$

By (43) with $K = 0$, since $\Delta_H^P \omega = \overline{\Delta}_H^P \omega = 0$, see (29), we get $\overline{\Delta}^P(\|\omega\|^2) \geq 0$. By Proposition 8 with $K = 0$ and $X = \nabla^P(\|\omega\|^2)$, we get $\Delta^P(\|\omega\|^2) = 0$. Applying Theorem 3(b), we get $\nabla^P \omega = 0$. $\quad\square$

Notice that, if $P(TM)$ in Theorems 5 and 6 is bracket-generating, then $\|\omega\| = \text{const}$ on $M$.

## 8. Appendix: The Almost Lie Algebroid Structure

Here, for the convenience of a reader, we briefly recall the construction of an almost Lie algebroid, following Section 2 in [13] (see also [15,16]). Lie algebroids (and Lie groupoids) constitute an active field of research in differential geometry. Roughly speaking, an (almost) Lie algebroid is a structure, where one replaces the tangent bundle $TM$ of a manifold $M$ with a new smooth vector bundle $\pi_{\mathcal{E}} : \mathcal{E} \to M$ of rank $k$ over $M$ (i.e., a smooth fiber bundle with fiber $\mathbb{R}^k$) with similar properties. Lie groupoids are related to Lie algebroids similarly as Lie groups are related to Lie algebras, see [24]. Lie algebroids deal with integrable distributions (foliations). Almost Lie algebroids are closely related to singular distributions, e.g., [13,14].

**Definition 7.** *An anchor on* $\mathcal{E}$ *is a morphism* $\rho : \mathcal{E} \to TM$ *of vector bundles. A skew-symmetric bracket on* $\mathcal{E}$ *is a map* $[\cdot, \cdot]_\rho : \mathfrak{X}_{\mathcal{E}} \times \mathfrak{X}_{\mathcal{E}} \to \mathfrak{X}_{\mathcal{E}}$ *such that*

$$[Y, X]_\rho = -[X, Y]_\rho, \qquad [X, fY]_\rho = \rho(X)(f)Y + f[X, Y]_\rho, \qquad \rho([X, Y]_\rho) = [\rho(X), \rho(Y)] \tag{46}$$

*for all* $X, Y \in \mathfrak{X}_{\mathcal{E}}$ *and* $f \in C^\infty(M)$. *The anchor and the skew-symmetric bracket give an almost Lie algebroid structure on* $\mathcal{E}$.

Note that axiom (46), third formula, is equivalent to vanishing of the following operator:

$$\mathfrak{D}^\rho(X, Y) = [\rho X, \rho Y] - \rho([X, Y]_\rho). \tag{47}$$

There is a bijection between almost Lie algebroids on $\mathcal{E}$ and the exterior differentials of the exterior algebra $\Lambda(\mathcal{E}) = \bigoplus_{k \in \mathbb{N}} \Lambda^k(\mathcal{E})$, [17]; here $\Lambda^k(\mathcal{E})$ is the set of $k$-forms over $\mathcal{E}$. The exterior differential $d^\rho$, corresponding to the almost Lie algebroid structure $(\mathcal{E}, \rho, [\cdot, \cdot]_\rho)$, is given by

$$
\begin{aligned}
d^\rho \omega(X_0, \dots, X_k) &= \sum_{i=0}^{k} (-1)^i (\rho X_i)(\omega(X_0, \dots, X_{i-1}, X_{i+1}, \dots, X_k)) \\
&\quad + \sum_{0 \leq i < j \leq k} (-1)^{i+j} \omega([X_i, X_j]_\rho, X_0, \dots, X_{i-1}, X_{i+1}, \dots, X_{j-1}, X_{j+1}, \dots, X_k),
\end{aligned}
$$

where $X_0, \dots, X_k \in \mathfrak{X}_{\mathcal{E}}$ and $\omega \in \Lambda^k(\mathcal{E})$ for $k \geq 0$. For $k = 0$, we have $d^\rho f(X) = (\rho X)(f)$, where $X \in \mathfrak{X}_{\mathcal{E}}$ and $f \in C^\infty(M) = \Lambda^0(\mathcal{E})$. Recall that a skew-symmetric bracket defines uniquely an exterior differential $d^\rho$ on $\Lambda(TM)$, and it gives rise to

- an *almost algebroid* if and only if $(d^\rho)^2 f = 0$ for $f \in C^\infty(M)$;

–   a *Lie algebroid* if and only if $(d^\rho)^2 f = 0$ and $(d^\rho)^2 \omega = 0$ for $f \in C^\infty(M)$ and $\omega \in \Lambda^1(TM)$.

**Definition 8.** A $\rho$-connection on $(\mathcal{E}, \rho)$ is a map $\nabla^\rho : \mathfrak{X}_\mathcal{E} \times \mathfrak{X}_\mathcal{E} \to \mathfrak{X}_\mathcal{E}$ satisfying Koszul conditions

$$\nabla^\rho_X (fY + Z) = \rho(X)(f)Y + f\nabla^\rho_X Y + \nabla^\rho_X Z, \qquad \nabla^\rho_{fX+Z} Y = f\nabla^\rho_X Y + \nabla^\rho_Z Y. \tag{48}$$

For a $\rho$-connection $\nabla^\rho$ on $\mathcal{E}$, they define the torsion $T^\rho : \mathfrak{X}_\mathcal{E} \times \mathfrak{X}_\mathcal{E} \to \mathfrak{X}_\mathcal{E}$ and the curvature $R^\rho : \mathfrak{X}_\mathcal{E} \times \mathfrak{X}_\mathcal{E} \times \mathfrak{X}_\mathcal{E} \to \mathfrak{X}_\mathcal{E}$ by standard formulas

$$T^\rho(X,Y) = \nabla^\rho_X Y - \nabla^\rho_Y X - [X,Y]_\rho, \qquad R^\rho_{X,Y} Z = \nabla^\rho_X \nabla^\rho_Y Z - \nabla^\rho_Y \nabla^\rho_X Z - \nabla^\rho_{[X,Y]_\rho} Z. \tag{49}$$

## 9. Conclusions

The main contribution of this paper is the further development of Bochner's technique for a regular or singular distribution parameterized by a smooth endomorphism $P$ of the tangent bundle of a Riemannian manifold with linear connection. In particular, the main results of this paper, Theorems 1–6 are proved. We introduce the concept of a statistical $P$-structure, i.e., a pair $(g, \overline{\nabla}^P)$ of a metric $g$ and $P$-connection $\overline{\nabla}^P$ on $M$ with a totally symmetric contorsion tensor $K$, see (10), and assume (13) for $P$ to use the concept of almost Lie algebroids. To generalize some geometrical analysis tools for distributions, we assume the additional conditions (21) and (35) for tensors $P$ and $K$. We introduce and study a Weitzenböck type curvature operator on tensors and prove vanishing theorems on the null space of the Hodge type Laplacian on a distribution with a statistical type connection.

We delegate the following for further study: (a) generalize some constructions in the paper, e.g., statistical $P$-structures, divergence results, to more general almost algebroids or Lie algebroids; (b) use less restrictive conditions on $K$; (c) find more applications in geometry and physics.

**Author Contributions:** Investigation, P.P., V.R. and S.S. All authors contributed equally in writing this article.

**Funding:** This research received no external funding.

**Conflicts of Interest:** The authors declare no conflict of interest.

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
