# Peer review of "On Singular Distributions With Statistical Structure"

_mathematics, doi:10.3390/math8101825_

Round 1

Reviewer 1 Report

The geometry of a Riemannian manifold $(M,g)$ endowed with a

singular or regular distribution is still being addressed because of its

various applications in the theoretical investigations [1,10,12,13,15-21,23,24] (numbering as in the submitted manuscript).

In the paper under review, the authors extend some previous studies

regarding a Riemannian manifold endowed with a singular (or regular)

distribution [16--20], generalizing Bochner's technique and a statistical

type connection. The main results are concentrated on the investigations of a modified statistical connection and then on some characterizations of the Weitzenb\"{o}ck type curvature operator on tensors (Theorem 4). Moreover, the Bochner--Weitzenb\"{o}ck type formula and vanishing theorems regarding the null space of the Hodge type Laplacian on a distribution (Theorems 5 and 6) are established.

The referee thinks that the main results in this paper are meaningful. Furthermore, the referee recommends this paper to be published in Mathematics.

Author Response

The revision of the article was done to show explicitly that we generalize the statistical structure and the Bochner technique for distributions. Some notations were changed to make compatibility with our previous paper [21], for example, symbols over connection, Laplacian, curvature tensor, etc.:

a) \overline is used for connection and objects that depend on K,

b) \smile for dual connection and objects, and connection,

c) objects without K are as in our previous paper.

Author Response

The revision of the article was done. Details are in the pdf file.

Reviewer 3 Report

In [Ann. Global Anal. Geom. 48 (2015), no. 4, 357–395], Opozda generalized Bochner's technique to statistical structures. In this paper, the authors extend Bochner’s technique to a Riemannian manifold equipped with a singular (or regular) distribution and a statistical type connection. They definine the Weitzenböck type curvature operator on tensors, prove the Bochner–Weitzenböck type formula and obtain some vanishing results. The given results are new, interesting, and well-motivated. Therefore, I recommend the article for publication. However, there are some minor corrections to be made.

  1. Page 1, line 30: after "...statistics", add “as well as in information geometry”.
  2. Page 2, after eq. (2), specify the meaning of $e_i$.
  3. Page 4, line 87: $c$ is not defined.
  4. Use the same notation for the Lie algebra of smooth vector fields on $M$. See, e.g., page 2, line 43 and page 6, line 117.
  5. Page 8, after eq. (26), specify the meaning of “^”.

Author Response

(The authors gave the same response as above.)

Round 2

Reviewer 2 Report

See attached report.
